# Impact of DTaP-IPV and DTaP Vaccination Among Adult Allogeneic Hematopoietic Stem Cell Transplant Recipients: A Prospective Observational Study

**DOI:** 10.3390/vaccines13030275

**Published:** 2025-03-05

**Authors:** Taiichiro Kobayashi, Sho Fujiwara, Ayako Ide, Takashi Toya, Naoki Shingai, Hiroaki Shimizu, Yuho Najima, Takeshi Kobayashi, Noriko Doki, Aoi Jo

**Affiliations:** 1Vaccine Clinic, Tokyo Metropolitan Cancer and Infectious Diseases Center Komagome Hospital, 3-18-22 Honkomagome, Bunkyo-ku, Tokyo 113-8677, Japan; fujiwarackn@gmail.com (S.F.); ayako_ide@tmhp.jp (A.I.); aoi_jou@tmhp.jp (A.J.); 2Department of Infectious Diseases, Tokyo Metropolitan Cancer and Infectious Diseases Center Komagome Hospital, 3-18-22 Honkomagome, Bunkyo-ku, Tokyo 113-8677, Japan; 3Department of Pediatrics, Tokyo Metropolitan Cancer and Infectious Diseases Center Komagome Hospital, 3-18-22 Honkomagome, Bunkyo-ku, Tokyo 113-8677, Japan; 4Hematology Division, Tokyo Metropolitan Cancer and Infectious Diseases Center Komagome Hospital, 3-18-22 Honkomagome, Bunkyo-ku, Tokyo 113-8677, Japan; tooya-tky@umin.ac.jp (T.T.); naoki_shingai@tmhp.jp (N.S.); hiroaki_shimizu@tmhp.jp (H.S.); yuuhou_najima@tmhp.jp (Y.N.); takepy@hotmail.com (T.K.); noriko_doki@tmhp.jp (N.D.)

**Keywords:** DTaP-IPV, DTaP, vaccines, adult, allogeneic hematopoietic stem cell transplant

## Abstract

**Background/Objectives**: Hematopoietic stem cell transplantation (HSCT) can potentially cure hematological malignancies; however, post-transplant patients have a high risk of infection owing to their immunocompromised status. Vaccination against pathogens, such as diphtheria, tetanus, pertussis, and polio, is essential post-transplantation, but neither the long-term efficacy of vaccines nor the optimal vaccination schedule has been fully established. **Methods**: In this prospective observational study, we assessed the short- and long-term immunogenicity of three doses of the diphtheria, tetanus, acellular pertussis, and inactivated poliovirus (DTaP-IPV) vaccines or DTaP vaccines in 29 adult allogeneic HSCT (allo-HSCT) recipients, with antibody levels measured at baseline, 1–3 months post-vaccination, and 1-year after vaccine completion. **Results**: At baseline, a substantial proportion of patients lacked protective antibody levels for the targeted pathogens. However, within 1–3 months post-vaccination, seropositivity rates significantly increased, reaching 78–100% for diphtheria, tetanus, pertussis, and poliovirus. Despite this, antibody levels significantly declined 1-year post-vaccination, especially for pertussis, with only 58–65% of patients maintaining protective levels. In contrast, 85–96% of patients retained protective levels for diphtheria, tetanus, and poliovirus, although antibody values also decreased. Compared to human leukocyte antigen (HLA)-mismatched cases, HLA-matched cases showed significantly higher antibody levels for diphtheria, pertussis, and poliovirus types 1 and 3. **Conclusions**: This study demonstrates the short-term effectiveness of DTaP-IPV and DTaP vaccines in adult allo-HSCT patients but emphasizes the challenge of maintaining long-term immunity. Given the difficulties in sustaining long-term vaccine efficacy in allo-HSCT recipients, particularly in HLA-mismatched cases, re-evaluating the current vaccination schedule may be necessary to maintain protection.

## 1. Introduction

Hematopoietic stem cell transplantation (HSCT) for leukemia and other hematological malignancies has steadily increased worldwide, with around 84,000 HSCTs performed annually, including 6000 in Japan [1,2]. Despite its curative potential, HSCT increases the risk of infection due to post-transplant compromised immunity and the use of immunosuppressive drugs for the prophylaxis and treatment of graft-versus-host disease (GVHD). As a result, post-transplant revaccination is essential to restore immunity and protect against vaccine-preventable diseases (VPDs). Revaccination against infectious diseases such as pneumococcus, *Haemophilus influenzae* type B (Hib), diphtheria, tetanus, pertussis, polio, hepatitis B, influenza, measles, rubella, mumps, and varicella is recommended post-transplantation, as per global guidelines [3,4,5,6]. Live vaccines, however, are contraindicated in immunocompromised recipients and are reserved for individuals who meet stringent safety criteria. These vaccines are also part of routine childhood immunization programs worldwide. This indicates that these infectious diseases are globally significant, either due to their severity or high incidence. Although the widespread implementation of vaccination programs has led to a global decline in tetanus, diphtheria, and poliomyelitis cases, a resurgence in pertussis has been observed in several countries. According to the World Health Organization (WHO), more than 151,000 cases of pertussis were reported worldwide in 2018, highlighting its ongoing public health burden.

Vaccination improves post-transplant outcomes. Nevertheless, gaps persist in optimal vaccination strategies, including formulations, schedules, and other related factors. Data on the clinical and immunological efficacy of diphtheria, tetanus, pertussis, and polio vaccines in adult allogeneic HSCT (allo-HSCT) patients remain limited. The timing of vaccine initiation varies widely, ranging from 6 months to over 1-year post-transplantation [7,8,9,10]. Vaccine formulations also differ, with diphtheria and tetanus toxoids and acellular pertussis adsorbed vaccine (DTaP) or tetanus toxoid and reduced diphtheria toxoid and acellular pertussis vaccine (Tdap) being used depending on the study [7,8]. In addition, clear guidelines on the optimal timing and necessity of additional vaccinations are lacking [3,4,5,6]. To date, most studies focus on short-term outcomes, such as antibody titers 1-month post-vaccination, with little investigation into long-term persistence of immunity. Despite three priming doses, the reported short-term seropositivity rates were relatively high: 95–96% for diphtheria, 96–100% for tetanus, 91% for pertussis, and 52% for poliovirus [7,8,9,10]. Nevertheless, concerns persist regarding the suboptimal seroconversion rates for poliovirus and the paucity of robust data concerning sustained vaccine-induced immunity, particularly for pertussis and poliovirus. This raises questions about the adequacy of current vaccination protocols in providing long-term immunity for adult allo-HSCT patients. Bridging these deficiencies is critical for enhancing patient prognosis.

Therefore, this study will prospectively evaluate the efficacy and durability of the diphtheria, tetanus toxoids, acellular pertussis adsorbed, and inactivated poliovirus vaccine (DTaP-IPV) or DTaP in adult post-allo-HSCT patients by assessing short- and long-term changes in antibody levels before and after vaccination, which will be initiated at least 6 months post-transplantation. Two priming doses will be given at 3–8 weeks apart, followed by a booster dose 6 months after the second dose. The European Society for Blood and Marrow Transplantation (EBMT) guidelines recommend three doses of DTaP and IPV doses at 4-week intervals starting at 6 months post-allo-HSCT, without a booster, raising concerns about long-term protection [3]. Conversely, the Infectious Diseases Society of America (IDSA) recommends three different schemes: (1) three doses of DTaP, (2) one dose of Tdap followed by two doses of DT, or (3) one dose of Tdap followed by two doses of Td, accompanied by a three-dose IPV series [4]. Nevertheless, the optimal timing between these doses is not detailed in IDSA’s recommendations. Additionally, the Centers for Disease Control and Prevention (CDC) recommends a three-dose series for unvaccinated adults: one dose of Tdap, followed by Td or Tdap after 4 weeks, and a third dose 6–12 months later [11]. Considering that DTaP and DTaP-IPV contain higher antigenic loads than Tdap or Td, neither of which is approved in Japan, they are preferable for eliciting enhanced immune responses. The Japanese post-HSCT guideline also recommends a three-dose schedule, although the specific intervals remain undefined, and both DTaP and DTaP-IPV are approved in Japan [12]. On another note, in many countries, post-HSCT vaccinations are self-funded, so reducing unnecessary vaccines is crucial. Given these factors, our institution has adopted this vaccination schedule and these formulations as part of routine clinical practice.

## 2. Materials and Methods

### 2.1. Study Design and Participants

This single-center prospective study evaluated the efficacy of vaccines against diphtheria, tetanus, pertussis, and polio in adult allo-HSCT recipients. The study was conducted at Tokyo Metropolitan Cancer and Infectious Diseases Center Komagome Hospital, a public 800-bed facility that performs over 120 allo-HSCTs annually. This study was approved by the Research Committee of Tokyo Metropolitan Cancer and Infectious Diseases Center Komagome Hospital. Written informed consent was obtained from all recipients prior to enrollment.

Vaccinations were administered as part of long-term follow-up (LTFU) outpatient care. As this study was designed as an observational study with a predefined enrollment period and was conducted within a limited budget, a formal sample size calculation was not performed. Enrollment occurred between March and May 2021. All clinically eligible patients for post-HSCT vaccination during this period were invited to participate. Eligible participants were adult patients scheduled for DTaP-IPV or DTaP vaccinations post-allo-HSCT, per the Japanese Society for Transplantation and Cellular Therapy (JSTCT) guideline. Inclusion criteria were age ≥20 years, relapse-free for ≥6 months post-transplantation, no progressive cGVHD, and written informed consent. Exclusion criteria included disease recurrence, severe cGVHD at enrollment, or being deemed unsuitable by the investigator. Unsuitability was defined as the presence of logistical barriers to study participation (e.g., inability to fully comprehend Japanese) or a poor prognosis that would preclude adequate follow-up. These were evaluated at the time of enrollment. The severity of GVHD was assessed according to the JSTCT guideline [13].

### 2.2. Vaccine

The vaccines used were DTaP-IPV (Quattrovac^®^, KM Biologics Co., Ltd., Kumamoto, Japan) and DTaP (Tribik^®^, the Research Foundation for Microbial Diseases of Osaka University, Osaka, Japan), both routinely used in Japan. The choice between DTaP-IPV and DTaP was made voluntarily by the patients. Those choosing DTaP did not request separate IPV. With reference to the prescribing information, the first two doses were planned to be administered 3 to 8 weeks apart, followed by an additional dose at least 6 months later.

### 2.3. Data Sampling and Antibody Measurement

Patient data were extracted from electronic medical records. Blood tests were conducted before vaccination, 1–3 months after completing the series, and 1-year post-vaccination to measure antibody levels for diphtheria, tetanus, pertussis, and poliovirus. Diphtheria antibodies were measured using the VaccZyme Anti-Diphtheria Toxoid IgG Enzyme Immunoassay Kit (Binding Site Ltd., Birmingham, UK); tetanus antibodies with the VaccZyme Anti-Tetanus Toxoid IgG Enzyme Immunoassay Kit (Binding Site Ltd., Birmingham, UK); pertussis antibodies with Pertussis EIA SEIKEN (Denka Co., Ltd., Niigata, Japan); and poliovirus antibodies with a neutralization assay (KM Biologics Co., Ltd., Kumamoto, Japan). Protective levels were ≥0.1 IU/mL for diphtheria and tetanus, ≥10 EU/mL for anti-PT (pertussis toxin) and anti-FHA (filamentous hemagglutinin) for pertussis, and a neutralizing titer of ≥1:8 for poliovirus [14,15,16]. A responder was defined as an individual whose antibody level increased fourfold or more [7,10].

### 2.4. Statistical Analysis

Patient characteristics were summarized using median (range) for continuous variables and percentage for categorical variables. A normality test was not performed due to the small sample size; however, geometric mean concentrations (GMCs) and titers (GMTs) with 95% confidence intervals (CIs), which are commonly used in vaccinology, were adopted. Due to the small sample size, a non-parametric test was used to ensure robust statistical comparisons. Antibody response and seropositivity were assessed with exact 95% binomial confidence intervals. GMCs and GMTs were calculated by assigning half the lower detection limit for values below the detection limit and diluting values exceeding the upper detection limit. Changes in GMCs and GMTs at pre-vaccination, 1–3 months post-vaccination, and 1-year post-vaccination were compared using the Wilcoxon matched-pairs signed rank test. The impact of various factors on GMCs and GMTs at each time point was assessed using the Mann–Whitney *U* or Kruskal–Wallis tests. Factors analyzed included age (≥50 years), sex, primary disease, human leukocyte antigen (HLA) matching level, stem cell source, maximum observed grade of GVHD, use of immunosuppressive agents (at vaccine initiation, at the third dose, and 1-year post-vaccination), IgG levels (≥500 mg/mL at vaccine initiation), lymphocyte count (≥2000/µL at vaccine initiation), vaccine type (DTaP-IPV or DTaP), and the time interval between HSCT and initiation of vaccination (≥9 months). Seropositivity rates for pertussis and other antigens were compared using McNemar’s test. All analyses used two-tailed *p*-values, with significance defined as *p* < 0.05. Missing data were handled via pairwise deletion, excluding cases from specific analyses where necessary. Participants receiving DTaP were excluded from poliovirus antibody titer analysis. Data were analyzed using Statistical Product and Service Solutions (SPSS, version 26.0, IBM, Chicago, IL, USA), GraphPad Prism (version 6.01, GraphPad Software, La Jolla, CA, USA), and G*Power (version 3.1.9.7, Heinrich-Heine-Universität, Düsseldorf, Germany).

## 3. Results

### 3.1. Study Population

From March to May 2021, all 30 post-allo-HSCT patients scheduled to receive vaccination were approached. However, 1 patient did not consent to participate, and 29 were enrolled. All 29 received the first and second vaccinations, but only 27 received the third, as two relapsed before the third dose. One additional patient relapsed after the third dose, leaving three patients unable to complete blood testing 1-year post-vaccination. No other patient transfers or visit interruptions occurred. In total, three patients withdrew from the study (Figure 1).

The median age was 53 years (range: 24–70), with 14 males (48%) and 15 females (52%). All were Japanese except one from Vietnam. Childhood vaccination records were unavailable. Among the 28 Japanese participants, all (100%) belonged to the generation routinely vaccinated against diphtheria, 14 (50%) against tetanus, all participants (100%) against pertussis, and 18 (64%) against polio. None had a history of these infectious diseases.

Acute myeloid leukemia (AML) was the primary disease in 18 patients (62%), acute lymphoblastic leukemia (ALL) in 5 (17%), myelodysplastic syndrome (MDS) in 4 (14%), and adult T-cell leukemia/lymphoma (ATL) and malignant lymphoma in 1 each (3%). Donor types were HLA-matched sibling (17%), matched unrelated (14%), mismatched sibling (3%), mismatched unrelated (45%), and haploidentical donors (21%). HSCT procedures included bone marrow transplantation (BMT) (48%), peripheral blood stem cell transplant (PBSCT) (39%), and cord blood transplant (CBT) (14%). After being informed about post-HSCT immunization, 27 patients (93%) chose DTaP-IPV and 2 (7%) selected DTaP; no patient switched vaccines. Among the recipients, five (17%) received the second vaccination within four weeks, and one (3%) received it one day earlier than the minimum three-week interval (Table 1).

### 3.2. Antibody Positivity Rates

Table 2 presents antibody response rates and seropositivity data. The median interval between transplantation and the first immunization was 257 days. Before vaccination, seropositivity was the following: diphtheria, 28% (95% CI, 10–45%); tetanus, 21% (95% CI, 5–36%); pertussis (PT), 31% (95% CI, 13–49%); pertussis (FHA), 21% (95% CI, 5–36%); poliovirus type 1, 59% (95% CI, 39–79%); poliovirus type 2, 89% (95% CI, 76–100%); and poliovirus type 3, 48% (95% CI, 28–68%). Many patients lacked sustained positive antibody levels post-allo-HSCT. The median interval between doses was 28 days (first to second) and 182 days (second to third). At 1–3 months post-vaccination, seropositivity was the following: diphtheria, 89% (95% CI, 79–100%); tetanus, 85% (95% CI, 71–100%); pertussis (PT), 81% (95% CI, 66–97%); pertussis (FHA), 78% (95% CI, 61–95%); poliovirus type 1, 92% (95% CI, 81–100%); poliovirus type 2, 100% (95% CI, 87–100%); and poliovirus type 3, 88% (95% CI, 74–100%). At 1-year post-vaccination, seropositivity was the following: diphtheria, 85% (95% CI, 70–99%); tetanus, 88% (95% CI, 75–100%); pertussis (PT), 58% (95% CI, 37–78%); pertussis (FHA), 65% (95% CI, 46–85%); poliovirus type 1, 88% (95% CI, 73–100%); poliovirus type 2, 96% (95% CI, 87–100%); and poliovirus type 3, 88% (95% CI, 73–100%).

Pertussis antibody (PT) is considered a more reliable key indicator of pertussis vaccine efficacy than pertussis antibody (FHA). Only the seropositivity rate for pertussis (PT) showed a significant decline between 1 and 3 months and 1 year after the third vaccination (*p* = 0.035, post hoc power: 87%). The decline in seropositivity rates at 1 year after vaccination for other pathogens did not reach statistical significance, with post hoc power analysis yielding a power of 99%. However, as no cases showed a transition from negative to positive, the estimated power is likely inflated. This issue is particularly relevant as multiple comparisons also resulted in a power of 99%, indicating potential overestimation across multiple tests. Binary data analysis identified a statistically significant difference only in the seropositivity rate for pertussis (PT), while numerical data in *Antibody Level Trends* revealed a significant decline in vaccine-induced antibody levels for all pathogens. The comparison of seropositivity rates for pertussis (PT) after vaccination at 1 year with those for other pathogens using McNemar’s test revealed that the seropositivity rate for pertussis (PT) was significantly lower in all comparisons (*p* < 0.05). Similarly, when compared with pertussis (FHA), the seropositivity rate for pertussis (FHA) was also significantly lower than those for tetanus and poliovirus type 2 (*p* < 0.05). Additionally, the seropositivity rate for pertussis (FHA) tended to be lower than those for the other antigens, with a trend toward statistical significance observed across all comparisons (*p* = 0.063) (Table 2).

### 3.3. Antibody Level Trends

Pre-vaccination GMCs and GMTs were the following: diphtheria, 0.04 IU/mL (95% CI, 0.02–0.08); tetanus, 0.06 IU/mL (95% CI, 0.05–0.08); pertussis (PT), 5.15 EU/mL (95% CI, 3.50–7.58); pertussis (FHA), 3.54 EU/mL (95% CI, 2.27–5.51); poliovirus type 1, 17.2 (95% CI, 9.0–33.0); poliovirus type 2, 36.4 (95% CI, 19.3–68.8); and poliovirus type 3, 10.7 (95% CI, 5.3–21.6). At 1–3 months post-vaccination, GMCs and GMTs were the following: diphtheria, 3.30 IU/mL (95% CI, 1.45–7.49; *p* < 0.001); tetanus, 1.37 IU/mL (95% CI, 0.75–2.50; *p* < 0.001); pertussis (PT), 53.00 EU/mL (95% CI, 29.02–96.78; *p* < 0.001); pertussis (FHA), 66.67 EU/mL (95% CI, 35.13–126.55; *p* < 0.001); poliovirus type 1, 224.5 (95% CI, 91.3–552.1; *p* < 0.001); poliovirus type 2, 2778.0 (95% CI, 1170.8–6591.8; *p* < 0.001); and poliovirus type 3, 1235.7 (95% CI, 417.2–3659.9; *p* < 0.001), indicating a significant increase compared to the pre-vaccination levels. At 1-year, GMCs and GMTs were the following: diphtheria, 0.59 IU/mL (95% CI, 0.28–1.24; *p* < 0.001); tetanus, 0.34 IU/mL (95% CI, 0.23–0.51; *p* < 0.001); pertussis (PT), 12.58 EU/mL (95% CI, 6.95–22.79; *p* < 0.001); pertussis (FHA), 15.99 EU/mL (95% CI, 9.14–27.97; *p* < 0.001); poliovirus type 1, 75.7 (95% CI, 35.7–160.2; *p* < 0.001); poliovirus type 2, 504.8 (95% CI, 253.1–1007.0; *p* < 0.001); and poliovirus type 3, 247.1 (95% CI, 103.1–592.4; *p* < 0.001), showing a significant decline compared to the 1–3 month levels (Figure 2).

Table 3 presents GMFRs (geometric mean fold rises) between each time points. For all vaccine antigens, GMCs and GMTs increased at 1–3 months compared to pre-vaccination and subsequently declined at 1 year. However, all GMCs and GMTs remained elevated at 1 year compared to pre-vaccination. The fact that GMFR 95% CIs did not span 1 suggests a significant change; however, this interpretation should be considered with caution due to the small sample size precluding formal normality testing (Table 3).

Univariate analysis assessed the effects of variables on antibody levels at each time points. By stratifying with HLA matching, HLA-matched cases had significantly higher antibody levels for diphtheria, pertussis (PT, FHA), and poliovirus types 1 and 3 at 1–3 months. This difference persisted at 1 year for diphtheria and pertussis (PT, FHA). Stratified analysis based on age, sex, primary disease, stem cell source, cGVHD severity, immunosuppressive use, initial IgG levels, lymphocyte counts, vaccine type, and time from HSCT to initiation of vaccination yielded a few statistically significant results, though these were considered incidental and are not detailed here (Figure 3).

### 3.4. Adverse Reactions

No serious adverse reactions occurred, except for mild fatigue and localized swelling at the injection site in a few patients.

## 4. Discussion

This study examined short- and long-term antibody changes after DTaP-IPV or DTaP vaccination in adult allo-HSCT recipients. Initially, many patients lacked protective antibody levels for diphtheria, tetanus, pertussis, and poliovirus. Within 1–3 months post-vaccination, significant antibody increases were seen, with 78–100% achieving protective levels for each pathogen. Nevertheless, by 1-year post-vaccination, antibody levels had significantly declined for all pathogens. Although diphtheria, tetanus, and poliovirus antibodies decreased, 85–96% remained seropositive. In contrast, only 58–65% retained protective pertussis antibodies, showing a sharper decline. The marked decline in antibody levels 1 year after vaccination, particularly for pertussis, was a novel finding in our study.

Previous studies also showed short-term antibody increases and protective levels after allo-HSCT vaccinations. For instance, 95–96% achieved protective levels against diphtheria, 98–100% for tetanus, 91% for pertussis, and 52% for poliovirus, consistent with our findings except for poliovirus. However, evidence on long-term antibody persistence is limited. Conrad et al. and Winkler et al. reported that, at approximately 1-year post-vaccination, 91–95% participants maintained protective antibodies for diphtheria, 100% for tetanus, 95% for pertussis, and 74% for poliovirus, though GMCs were not provided. Notably, they did not indicate a decline in antibody levels one-year post-vaccination [9,10].

In our study, we adopted a vaccination schedule consisting of two priming doses administered 3–8 weeks apart, followed by a booster dose at least 6 months later. In contrast, Conrad et al. used three priming doses 1 month apart, followed by a booster 1 year later, and Winkler et al. administered three doses 4 weeks apart, without a booster [9,10]. Differences in antibody levels across studies may reflect variations in vaccination schedules. However, vaccine efficacy also depends on the patient’s immune status. Indeed, studies suggest that vaccine initiation timing, immunosuppressive drugs, and cGVHD affect vaccine efficacy [17]. Conrad et al. initiated vaccination at 12.4 months (IQR, 10–18.4) post-HSCT, with 36.8% participants on immunosuppressive treatment and 33% participants with cGVHD [9]. On the other hand, Winkler et al. began vaccination at 226 days (range, 180–430) post-HSCT, with 69% participants on immunosuppressive therapy and 37% participants with moderate/severe cGVHD [10]. In our study, vaccine initiation times were similar to or earlier than those previously reported, cGVHD rates were similar to or higher, and the immunosuppressive drugs were administered in a high proportion, reaching 86%. Thus, our patients may have been less likely to achieve optimal vaccine responses. Based on the differences in vaccine schedules and patients’ immune status, our study confirmed that vaccination in adult post-allo-HSCT patients is effective in the short term; however, the long-term persistence of antibody levels, especially for pertussis, remains a challenge. Thus, regular antibody monitoring and reconsideration of booster timing may be needed, compared to the routine 10-year booster schedule for the general population [18]. Another analogous research is that antibody levels to pertussis and poliovirus do not persist for 10 years after immunization with DTaP-IPV or DTaP at age 0–1 years of age, even though it was initially thought that booster doses would only be required at age 11. As a result, an additional booster is recommended to be administered at age 5–6 years [19]. Our study raises the possibility that variations in the vaccination schedule and the immune status of the target population could influence the long-term maintenance of antibody titers. However, our findings do not provide definitive evidence to support a specific schedule modification, and further research is needed to clarify its potential impact. Given these uncertainties, future studies should focus on collecting long-term data beyond 1 year to better understand the durability of vaccine-induced immunity. Considering the need for boosters and possible extra primary doses, addressing the public health need for subsidizing vaccine costs for allo-HSCT and other immunosuppressed individuals might be important.

We observed that HLA-matched recipients showed a significantly greater antibody increase than HLA-mismatched recipients. This may be due to smoother immune reconstitution in HLA-matched transplants. HLA-mismatched transplants have a higher cGVHD risk, often requiring immunosuppressants, which may reduce vaccine response [9,20]. While we found no direct effects of cGVHD or immunosuppressants on vaccine response, their synergistic interaction might contribute to the statistically significant greater antibody rise in HLA-matched cases.

This single-center study involved 29 patients. Though the sample size is small, the good antibody response despite early vaccination, high cGVHD rates, and frequent immunosuppressant use is encouraging. Our study did not identify an association between the stem cell source (BMT, PBSCT, CBT) essential for immune reconstitution and the increase in antibody titers. Similarly, no correlation was observed between the presence of GVHD, which affects cellular immunity, and antibody responses. Given that such factors have been reported to influence antibody titers in response to other vaccines, they may also impact the immunogenicity of DTaP-IPV and DTaP [20]. However, the small sample size of this study may have limited our ability to detect such associations. Larger-scale studies are needed to improve generalizability and apply these findings to a wider range of patients. Furthermore, while we noted significant fluctuations in 1-year antibody positivity, long-term durability data are still lacking. Given the 10-year booster schedule for diphtheria, tetanus, and pertussis, prolonged follow-up is needed to assess immune durability. Lastly, this study not only had a small sample size but also exhibited considerable diversity within the analyzed group. Conversely, the majority of participants were Japanese. This diversity, along with genetic and geographical variations, must be taken into account when interpreting the findings. Further studies across diverse populations are warranted.

## 5. Conclusions

Our study demonstrated that DTaP-IPV and DTaP vaccines provide effective short-term protection in adult post-allo-HSCT patients, but maintaining long-term immunity, especially against pertussis, remains a challenge. While our findings do not definitively establish the need for a revised vaccination schedule, they suggest the possibility that adjustments in vaccination timing and frequency could be considered as a reasonable approach to mitigating antibody decline. Further research is needed to gather long-term data and explore potential refinements to vaccination schedules for optimal protection. These efforts are crucial for guiding clinical practice and public health vaccination policies.

## Figures and Tables

**Figure 1 vaccines-13-00275-f001:**
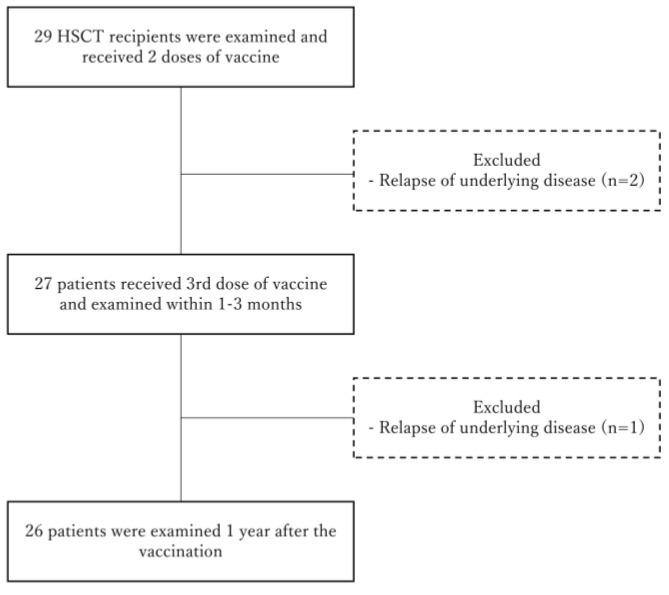
Flowchart of participants following enrollment. Abbreviation: HSCT, hematopoietic stem cell transplantation.

**Figure 2 vaccines-13-00275-f002:**
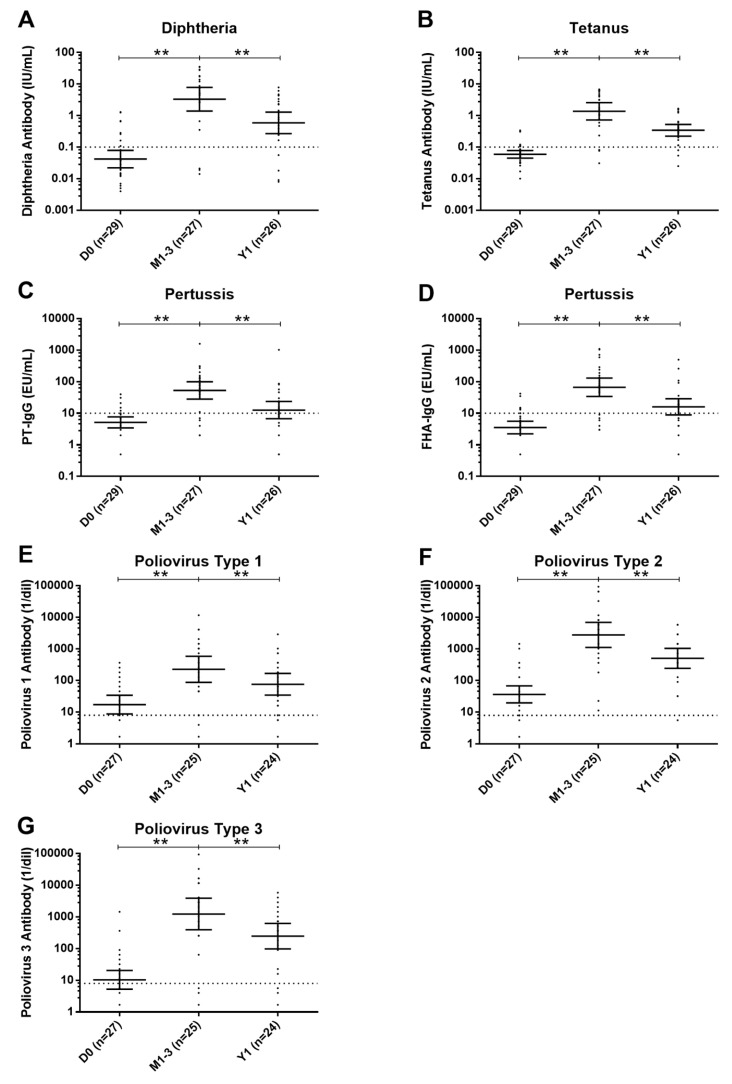
Changes in antibody levels for various pathogens. This figure illustrated the geometric mean concentrations/titers and 95% confidence intervals for specific antibodies: diphtheria (panel (**A**)), tetanus (panel (**B**)), pertussis toxin (PT) of *Bordetella pertussis* (panel (**C**)), filamentous hemagglutinin (FHA) of *B. pertussis* (panel (**D**)), poliovirus type 1 (panel (**E**)), poliovirus type 2 (panel (**F**)), and poliovirus type 3 (panel (**G**)). Measurements were taken before vaccination (D0), 1–3 months post-third vaccination (M1–3), and 1-year post-third vaccination (Y1). The horizontal dashed lines indicated the seroprotective antibody levels. Antibody concentrations/titers at each time points were compared using the Wilcoxon matched-pairs signed rank test. The level of statistical significance (*p* < 0.001) was indicated by the use of two asterisks (**). Abbreviations: D0, before vaccination; M1–3, 1–3 months after completion of vaccination; Y1, 1 year after completion of vaccination; PT, pertussis toxin; FHA, filamentous haemagglutinin.

**Figure 3 vaccines-13-00275-f003:**
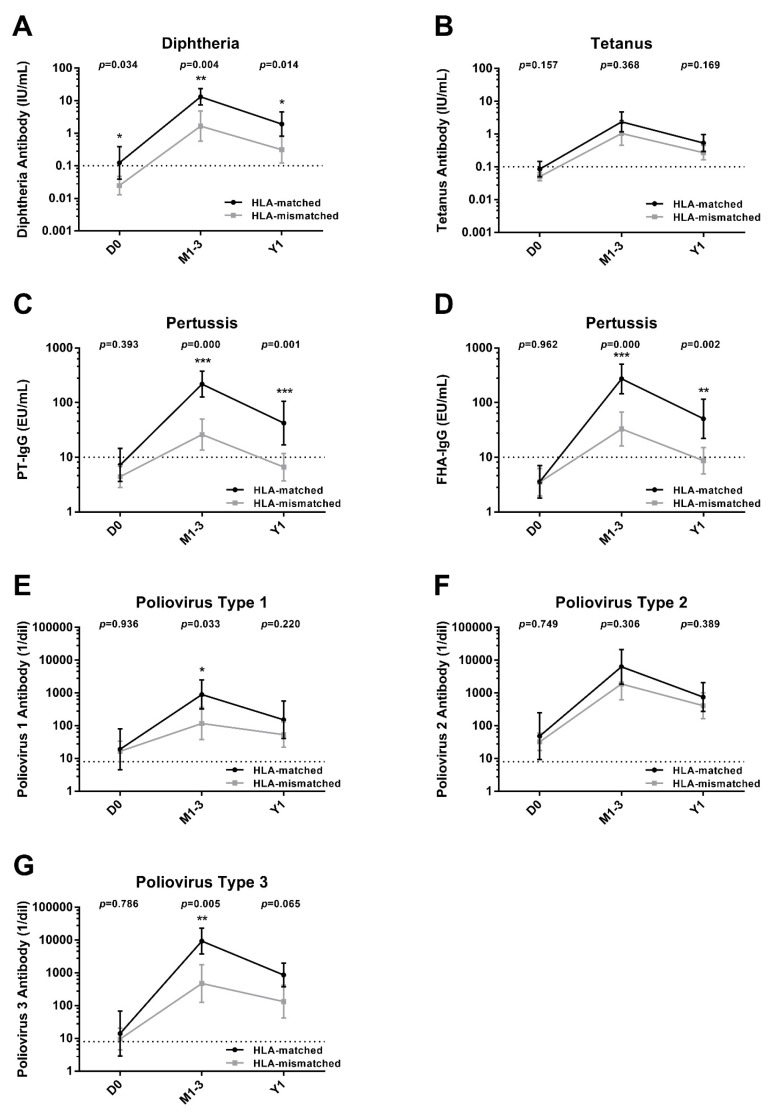
Changes in antibody levels for various pathogens stratified by HLA matching level (matched vs. mismatched). This figure illustrated the geometric mean concentrations/titers and 95% confidence intervals for specific antibodies: diphtheria (panel (**A**)), tetanus (panel (**B**)), pertussis toxin (PT) of *Bordetella pertussis* (panel (**C**)), filamentous hemagglutinin (FHA) of *B. pertussis* (panel (**D**)), poliovirus type 1 (panel (**E**)), poliovirus type 2 (panel (**F**)), and poliovirus type 3 (panel (**G**)). Measurements were taken before vaccination (D0), 1–3 months post-third vaccination (M1–3), and 1-year post-third vaccination (Y1). The data were stratified by human leukocyte antigen matching levels. The horizontal dashed lines indicated the seroprotective antibody levels. Antibody concentrations/titers at each time points were compared between matched and mismatched cases using the Mann–Whitney *U* test. The levels of statistical significance were indicated by asterisks as follows: *p* < 0.05 was denoted by one asterisk (*), *p* < 0.01 by two asterisks (**), and *p* < 0.001 by three asterisks (***). Abbreviations: D0, before vaccination; M1–3, 1–3 months after completion of vaccination; Y1, 1 year after completion of vaccination; HLA, human leukocyte antigen; PT, pertussis toxin; FHA, filamentous haemagglutinin.

**Table 1 vaccines-13-00275-t001:** Characteristics of study participants.

Characteristics	No. (%) or Median (Range)
Number of patients	29
Age, years	53 (24–70)
Male sex	14 (48)
Japanese	28 (97)
Underlying disease	
AML	18 (62)
ALL	5 (17)
MDS	4 (14)
ATL	1 (3)
ML	1 (3)
Donor type	
HLA-matched sibling donor	5 (17)
HLA-matched unrelated donor	4 (14)
HLA-mismatched sibling donor	1 (3)
HLA-mismatched unrelated donor	13 (45)
HLA-haploidentical donor	6 (21)
Stem cell source	
BMT	14 (48)
PBSCT	11 (39)
CBT	4 (14)
Conditioning regimen	
Myeloablative conditioning	18 (62)
Reduced-intensity conditioning	11 (38)
Maximum prior grade of acute GVHD	
No/grade I	17 (59)
Grade II-IV	12 (41)
Maximum prior grade of chronic GVHD	
No/mild	18 (62)
Moderate/severe	11 (38)
Immunosuppressant at vaccine initiation	25 (86)
Tacrolimus	16 (55)
Cyclophosphamide	5 (17)
Prednisolone	15 (52)
Others	6 (21)
No immunosuppressant	4 (14)
Rituximab administration	0 (0)
IVIg during vaccine period	3 (10)
IgG, mg/dL	712 (328–1822)
White blood cell count, /µL	5600 (1600–11,500)
Lymphocyte counts, /µL	1940 (290–4160)
Vaccine type	
DTaP-IPV	27 (93)
DTaP	2 (7)
Days of vaccination after HSCT	257 (182–2857)
Days between first and second vaccination	28 (20–77)
Days between second and third vaccination	182 (175–224)

Abbreviations: AML, acute myeloblastic leukemia; ALL, acute lymphoblastic leukemia; MDS, myelodysplastic syndrome; ATL, adult T-cell leukemia-lymphoma; ML, malignant lymphoma; HLA, human leukocyte antigen; BMT, bone marrow transplantation; PBSCT, peripheral blood stem cell transplantation; CBT, cord blood transplantation; GVHD, graft-versus-host disease; IVIg, intravenous immunoglobulin; IgG, immunoglobulin G; DTaP, diphtheria–tetanus–pertussis vaccine; IPV, inactivated polio vaccine; HSCT, hematopoietic stem cell transplantation.

**Table 2 vaccines-13-00275-t002:** Antibody response rate and seropositivity at each time points of the study.

Timing of Examination	Before Vaccine Series	1–3 Months After Third Dose	1 Year After Third Dose	1–3 Months vs. 1 Year, *p*
Diphtheria				
Responder	-	23/27 (85% [73–100])	18/26 (69% [50–88])	0.031 (99%)
Seroprotection	8/29 (28% [10–45])	24/27 (89% [79–100])	22/26 (85% [70–99])	0.500 (99%)
Tetanus				
Responder	-	23/27 (85% [71–100])	16/26 (62% [41–82])	0.035 (87%)
Seroprotection	6/29 (21% [5–36])	23/27 (85% [71–100])	23/26 (88% [75–100])	0.500 (99%)
Pertussis (PT)				
Responder	-	22/27 (81% [66–97])	8/26 (31% [12–50])	0.000 (99%)
Seroprotection	9/29 (31% [13–49])	22/27 (81% [66–97])	15/26 (58% [37–78])	0.035 (87%)
Pertussis (FHA)				
Responder	-	22/27 (81% [66–97])	16/26 (62% [41–82])	0.031 (99%)
Seroprotection	6/29 (21% [5–36])	21/27 (78% [61–95])	17/26 (65% [46–85])	0.125 (99%)
Poliovirus type 1				
Responder	-	14/25 (56% [35–77])	9/24 (38% [17–58])	0.063 (99%)
Seroprotection	16/27 (59% [39–79])	23/25 (92% [81–100])	21/24 (88% [73–100])	0.500 (99%)
Poliovirus type 2				
Responder	-	21/25 (84% [69–99])	17/24 (71% [51–90])	0.125 (99%)
Seroprotection	24/27 (89% [76–100])	25/25 (100% [87–100])	23/24 (96% [87–100])	NA (NA)
Poliovirus type 3				
Responder	-	20/25 (80% [63–97])	18/24 (75% [56–94])	0.500 (99%)
Seroprotection	13/27 (48% [28–68])	22/25 (88% [74–100])	21/24 (88% [73–100])	NA (NA)

The proportions of responders and seropositivity at 1–3 months and 1 year after the third dose, along with 95% confidence intervals, were compared using the directional McNemar test, with *p*-values and post hoc power values also provided. Abbreviations: PT, pertussis toxin; FHA, filamentous hemagglutinin; NA, not assessed.

**Table 3 vaccines-13-00275-t003:** Geometric mean fold rise in antibody levels following vaccination.

Characteristics	M1–3/D0	Y1/M1–3	Y1/D0
	n = 25–27	n = 24–26	n = 24–26
Diphtheria	74.91 (32.83–170.95)	0.18 (0.13–0.25)	15.16 (6.98–32.93)
Tetanus	22.73 (12.35–41.84)	0.27 (0.19–0.37)	5.82 (3.76–9.01)
Pertussis PT	10.92 (5.87–20.34)	0.24 (0.18–0.32)	2.69 (1.53–4.72)
Pertussis FHA	20.23 (10.12–40.42)	0.24 (0.18–0.32)	5.02 (2.77–9.08)
Poliovirus 1	12.26 (4.11–36.58)	0.34 (0.23–0.52)	4.41 (1.66–11.73)
Poliovirus 2	75.50 (27.44–207.72)	0.18 (0.12–0.27)	14.88 (6.58–33.65)
Poliovirus 3	113.41 (36.43–353.06)	0.20 (0.12–0.34)	24.77 (10.07–60.91)

Geometric mean fold rises (GMFRs) and 95% confidence intervals were calculated for antibody levels at M1–3 and Y1: GMFR of M1–3 to D0 (M1–3/D0), GMFR of Y1 to M1–3 (Y1/M1–3), and GMFR of Y1 to D0 (Y1/D0). Abbreviations: D0, before vaccination; M1–3, 1–3 months after completion of vaccination; Y1, 1 year after completion of vaccination; PT, pertussis toxin; FHA, filamentous haemagglutinin.

## Data Availability

The data supporting the findings of this study are not publicly available due to privacy and ethical restrictions.

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
