# Peer review of "Impact of DTaP-IPV and DTaP Vaccination Among Adult Allogeneic Hematopoietic Stem Cell Transplant Recipients: A Prospective Observational Study"

_vaccines, 2025, doi:10.3390/vaccines13030275_

Round 1

Reviewer 1 Report

Comments and Suggestions for Authors

Dear Authors,

The topic is essential and current, especially in the face of the increasing number of pertussis infections. The abstract is clear. The introduction satisfactorily explains the context and aims of the work. Materials and methods require additions. This study appears to be conducted reliably.

The analyzed group is diverse regarding diagnosis, type of transplanted hematopoietic cells, and type of immunosuppressive treatment used during vaccination. The vaccination schedule differs in the interval between the administered doses. Considering the multi-stage course of immune reconstitution, these factors may have affected the obtained vaccination results.

However, the results confirmed the humoral functional deficiencies observed in HSCT recipients, including remarkably shortened humoral response to pertussis vaccination.

The discussion should be updated, and the tables require corrections.

To conclude, this study requires major revision.

My suggestions are as follows

 Line 42

Please add “other.”

For leukemia and other haematological malignancies

Line 46

Please correct and add “treatment.”

 prophylaxis and treatment of GvHD

Line 48

The use of live vaccines is possible only in immunocompetent recipients (after meeting strictly defined conditions). The current description may be misleading for someone unfamiliar with the principles of post-transplant care.

Line 72 (and line 107)

The recommended interval between two first vaccine doses is at least 4 weeks (according to the CDC). Please specify how many patients were vaccinated earlier than 4 weeks.

Line 100

Please specify what criteria were used to assess the severity of GvHD.

Line 102

Please explain when the recipient was unsuitable by the investigator.

Line 120

Please explain why this antibody level was used in the responder definition.

Line 133

Why did analyzed factors include cGvHD ≥ moderate, while inclusion criteria excluded severe GvhD?

Table 1

Is it an acute GvHD diagnosed earlier and given in the history or a late onset?

Chronic GvhD- as above, in a comment to line 133

There is no information about the type of conditioning used

Please correct the dot instead of the comma in days of vaccination after HSCT

The text stated that the interval between the first and second vaccination was a maximum of 8 weeks ( 77 days is higher).

Please provide the total number of patients treated with immunosuppressive therapy ( in the discussion was as high as 86%)

Line 233

Please provide the results of the univariate analysis assessing the effect of HLA matching and lymphocyte count on antibody levels.

Discussion

It is worth discussing the issue of the source of hematopoietic cells used in transplantation (BMT, PBSCT, CBT), which is essential for the immune reconstitution.

Please consider the diversity of the analyzed group and the accompanying disorders of cellular immunity observed after HSCT (especially in GvHD patients), which also may have influenced the obtained results.

Author Response

Response to Reviewers

Manuscript Title: Impact of DTaP-IPV and DTaP Vaccination among Adult Allogeneic Hematopoietic Stem Cell Transplant Recipients: A Prospective Cohort Study

Manuscript ID: vaccines-3475396

Dear Reviewer,

We sincerely appreciate the reviewer’s insightful comments and constructive feedback. We have carefully addressed each point and made revisions to improve the clarity and scientific rigor of the manuscript.

We have provided a detailed point-by-point response below, outlining the revisions made. We kindly ask for your review of these changes and would greatly appreciate any further guidance on areas that may require additional refinement.

Point-by-Point Response:

Line 42:

Reviewer’s Comment: Please add “other.” For leukemia and other haematological malignancies.

Response: Thank you for your suggestion. We have added “other” as requested. (Line 43 in revised manuscript).

Line 46:

Reviewer’s Comment: Please correct and add “treatment.” prophylaxis and treatment of GvHD.

Response: We have revised the phrase to " prophylaxis and treatment of graft-versus-host-disease (GVHD)" as suggested. (Line 47 in revised manuscript).

Line 48:

Reviewer’s Comment: The use of live vaccines is possible only in immunocompetent recipients (after meeting strictly defined conditions). The current description may be misleading for someone unfamiliar with the principles of post-transplant care.

Response: We have clarified this statement to emphasize that live vaccines are only administered in immunocompetent recipients who meet strict criteria. (Line 52 in revised manuscript).

Line 72 & 107:

Reviewer’s Comment: The recommended interval between two first vaccine doses is at least 4 weeks (according to the CDC). Please specify how many patients were vaccinated earlier than 4 weeks.

Response: We have added the number of patients who received vaccinations earlier than 4 weeks in the Results section of the revised manuscript (Line 194 in revised manuscript).

Line 100:

Reviewer’s Comment: Please specify what criteria were used to assess the severity of GvHD.

Response: We have added details on the criteria used for GVHD severity assessment, following the Japanese Society for Transplantation and Cellular Therapy (JSTCT) guideline. (Line 122 in revised manuscript).

Line 102:

Reviewer’s Comment: Please explain when the recipient was unsuitable by the investigator.

Response: We have included a clarification in the Methods and Materials section. (Line 120 in revised manuscript).

Line 120:

Reviewer’s Comment: Please explain why this antibody level was used in the responder definition.

Response: We referred to previous studies and have added the relevant citation to support our definition. Additionally, a four-fold increase has historically been used in both research and clinical practice. (Line 143 in revised manuscript).

Line 133:

Reviewer’s Comment: Why did analyzed factors include cGvHD ≥ moderate, while inclusion criteria excluded severe GvHD?

Response: Patients with worsening or severe cGVHD at the time of enrollment were excluded. However, those with cGVHD were included if their cGVHD was stable and they were eligible for vaccination. Table 1 describes the highest severity of cGVHD observed before the study period (Line 119 in the revised manuscript).

Table 1:

Reviewer’s Comment 1: Is it an acute GvHD diagnosed earlier and given in the history or a late onset?

Response: It refers to acute GVHD diagnosed earlier and recorded in the history. During the study period, a considerable amount of time had passed since HSCT, and no patients presented with aGVHD. To avoid misunderstanding, we have revised the description accordingly (Table 1 in the revised manuscript).

Reviewer’s Comment 2: Chronic GvHD- as above, in a comment to line 133.

Response: Table 1 describes the highest severity of cGVHD observed before the study period (Table 1 in the revised manuscript).

Reviewer’s Comment 3: There is no information about the type of conditioning used.

Response: We have added this information (Table 1 in the revised manuscript).

Reviewer’s Comment 4: Please correct the dot instead of the comma in days of vaccination after HSCT.

Response: We have corrected two thousand eight hundred fifty-seven as instructed (Table 1 in the revised manuscript).

Reviewer’s Comment 5: The text stated that the interval between the first and second vaccination was a maximum of 8 weeks (77 days is higher).

Response: The vaccine prescribing information states an interval of 3 to 8 weeks; however, since this study is an observational study rather than an interventional trial, deviations in real-world clinical practice are expected. We have added the clarification (Line 128 in the revised manuscript).

Reviewer’s Comment 6: Please provide the total number of patients treated with immunosuppressive therapy (in the discussion was as high as 86%).

Response: This information has been included in Table 1 from the initial submission. "Immunosuppressant at vaccine initiation" was recorded for 25 patients (86%), and the breakdown is also provided. Please review the details in Table 1.

Line 233:

Reviewer’s Comment: Please provide the results of the univariate analysis assessing the effect of HLA matching and lymphocyte count on antibody levels.

Response: The original Figure 3 already illustrates the longitudinal changes in antibody levels stratified by HLA matching. Within Figure 3, we performed a univariate analysis at each time point to compare antibody levels between HLA-matched and HLA-mismatched cases, with p-values provided. Statistically significant differences are indicated by asterisks. We believe that stratification by HLA-matching provides meaningful insights. Regarding lymphocyte counts, similar to many other variables analyzed (age, sex, primary disease, stem cell source, cGVHD severity, immunosuppressive use, initial IgG levels, vaccine type, and time from HSCT to vaccination), no significant findings were observed. Given the lack of meaningful results and the extensive amount of additional data that would be required the extensive space required to present one-page-per-variable data as HLA matching, we carefully considered the balance between comprehensiveness and clarity and decided to focus on the most relevant results. This decision is already stated in the Results section. We appreciate your understanding (Line 273 and Figure 3 in the revised manuscript).

Discussion:

Reviewer’s Comment: It is worth discussing the issue of the source of hematopoietic cells used in transplantation (BMT, PBSCT, CBT), which is essential for the immune reconstitution. Please consider the diversity of the analyzed group and the accompanying disorders of cellular immunity observed after HSCT (especially in GvHD patients), which also may have influenced the obtained results.

Response: We have expanded the Discussion section to address the impact of different hematopoietic stem cell sources on immune reconstitution and included considerations regarding diversity within the analyzed group and cellular immunity disorders, particularly in patients with GVHD (Line 351 and 362 in revised manuscript).

Conclusion:

We appreciate the reviewer’s detailed feedback, which has greatly improved our manuscript. We have carefully addressed all comments and made the necessary revisions accordingly. We hope that the revised manuscript meets your expectations and look forward to your further evaluation. We would greatly appreciate any further guidance on areas that may require additional refinement.

Sincerely,

Taiichiro Kobayashi, M.D., Ph.D. (taiichirou_kobayashi@tmhp.jp)

Department of Infectious Diseases

Tokyo Metropolitan Cancer and Infectious Diseases Center Komagome Hospital

3-18-22, Honkomagome, Bunkyo-ku, Tokyo 113-8677, Japan

Reviewer 2 Report

Comments and Suggestions for Authors

1) Please add information about the Ethic Committee approval. 

2) Please add information about the sample size and the sample size calculation. Why did the authors study 29 patients?

3) What was the criteria to select DTaP-IPV or DTaP?

4) If it was a prospective study, at which moment did the patient sign the informed consent?

5) Why did the authors report the geometric median titers? They employed a non-parametric statistical test.

6) How did the authors exclude a bias of the results in the three patients that received DTaP and not DTaP-IPV? Are the immunogenicities of both vaccines similar (according to prior studies)?

7) Please perform a chi-square or exact Fisher test on the data of Table 2. It is clear that between basal and 1-3 measures, this was statistically significant. But what about the 1-3 measure vs 1 year? For example, PT responders in the 1-3 measure vs. 1 year, there is a statistically significant difference.

8) How did the authors explain the result of the example in point seven?

9) Which statistical test was performed to determine the data distribution of the information in figure 2? Data distribution seems to be normal, but the authors performed a non-parametric test. Why?

10) Data from figure 2 shows a statistically significant difference between M1-3 vs Y1, but not in seropositivity. Please make a deeper interpretation. Data from figure 3 seems not to be congruent with data from table 2.

11) For Table 3, please provide the statistical analysis to determine if the differences were or were not significant. 

12) Why is the graphic of figure 2 different from figure 3?

13) Why, if in figure 3 they employed a graphic that assumes a parametric distribution, did the authors perform a non-parametric test?

14) Also, data from time point 0 is related to point 1 and point 2, but the authors employed a statistical test to un-relate data (independent observation). This is not correct; they should employ a statistical test for dependent or related observations; they are the same patients at a different point in time. This applies for all the analysis.

Author Response

Response to Reviewers

Manuscript Title: Impact of DTaP-IPV and DTaP Vaccination among Adult Allogeneic Hematopoietic Stem Cell Transplant Recipients: A Prospective Cohort Study

Manuscript ID: vaccines-3475396

Dear Reviewer,

We sincerely appreciate the reviewer’s insightful comments and constructive feedback. We have carefully addressed each point and made revisions to improve the clarity and scientific rigor of the manuscript.

We have provided a detailed point-by-point response below, outlining the revisions made. We kindly ask for your review of these changes and would greatly appreciate any further guidance on areas that may require additional refinement.

Point-by-Point Response:

Reviewer’s Comment: Please add information about the Ethic Committee approval.

Response: Thank you for your suggestion. This information was already included in the Institutional Review Board Statement at the end of the initial manuscript. However, we have also added it to the Materials and Methods section for clarity (Line 106 and 382 in the revised manuscript).

Reviewer’s Comment: Please add information about the sample size and the sample size calculation. Why did the authors study 29 patients?

Response: We did not perform a sample size calculation as we did not have the financial resources to enroll a larger number of patients. During the pre-specified enrollment period, all 30 patients who were clinically eligible for post-HSCT vaccination were invited to participate, and 29 consented. (Line 111 in the revised manuscript).

Reviewer’s Comment: What was the criteria to select DTaP-IPV or DTaP?

Response: The choice between DTaP-IPV and DTaP was made voluntarily by the patients. We have added this clarification in the manuscript. (Line 127 in the revised manuscript).

Reviewer’s Comment: If it was a prospective study, at which moment did the patient sign the informed consent?

Response: Written informed consent was obtained at enrollment. We have added this clarification in the manuscript. (Line 108 in the revised manuscript).

Reviewer’s Comment: Why did the authors report the geometric median titers? They employed a non-parametric statistical test.

Response: We appreciate the reviewer's insightful comment. While geometric mean titers (GMTs) are typically used in conjunction with parametric tests, we reported GMTs as they are a widely accepted measure for summarizing antibody titers in vaccine studies, even when dealing with small sample sizes. Given that our study included only 29 patients, formal normality testing was not conducted due to the limited sample size. However, since small sample distributions are often skewed and challenging to assess for normality, we employed a non-parametric statistical test to ensure robustness in statistical comparisons. This approach has been widely adopted in numerous vaccine immunogenicity studies with limited sample size conducted to date. We deeply appreciate your understanding.

Reviewer’s Comment: How did the authors exclude a bias of the results in the three patients that received DTaP and not DTaP-IPV? Are the immunogenicity of both vaccines similar (according to prior studies)?

Response: In this study, both DTaP and DTaP-IPV vaccines were used, as they are the most commonly administered vaccines for routine immunization and post-HSCT revaccination in Japan. While no direct comparative studies exist, previous research has shown that both vaccines achieve nearly 100% seropositivity rates with comparable increases in antibody titers. To assess potential differences between the two vaccines, we conducted a univariate analysis comparing post-vaccination antibody titers between the two groups. We have added a statement indicating that no significant differences in antibody titers were observed between DTaP and DTaP-IPV (Line 161 and 273 in the revised manuscript).

Reviewer’s Comment: Please perform a chi-square or exact Fisher test on the data of Table 2. It is clear that between basal and 1-3 measures, this was statistically significant. But what about the 1-3 measure vs 1 year? For example, PT responders in the 1-3 measure vs. 1 year, there is a statistically significant difference.

Response: We have added the requested chi-square or Fisher’s exact test results to Table 2 to assess statistical significance between the 1–3 month and 1-year measurements. (Table 2 in the revised manuscript).

Reviewer’s Comment: How did the authors explain the result of the example in point seven?

Response: In the binary data analysis of seropositivity status (positive or negative), a statistically significant decline in positivity rates was rarely observed due to the limited data resolution. However, in Figure 2, where quantitative antibody titers were analyzed, all measurements showed a significant decline. For example, even if seropositivity was barely maintained one year after vaccination, the antibody titers had significantly decreased. We have added this result to the Results section. (Line 219 in the revised manuscript).

Reviewer’s Comment: Which statistical test was performed to determine the data distribution of the information in figure 2? Data distribution seems to be normal, but the authors performed a non-parametric test. Why?

Response: Given that our study included only 29 patients, we did not perform a formal normality test, as small sample sizes often limit the power of such tests to reliably assess distributional assumptions. Therefore, we opted for a non-parametric statistical test to ensure robustness in statistical comparisons. Furthermore, previous vaccine studies with limited sample sizes have also reported antibody titers using GMTs while employing non-parametric tests for statistical analysis, which we understand to be a well-accepted approach in vaccinology. We have added this clarification to the Methods section. We hope this clarification addresses your concern (Line 146 in the revised manuscript).

Reviewer’s Comment: Data from figure 2 shows a statistically significant difference between M1-3 vs Y1, but not in seropositivity. Please make a deeper interpretation. Data from figure 3 seems not to be congruent with data from table 2.

Response: Apologies for the redundancy with our previous response. The difference in statistical significance between the analyses likely stems from the differences in analytical approaches between binary and continuous data. Seropositivity (binary data) was derived from antibody titers by applying a cutoff threshold, which resulted in some loss of information and reduced statistical power. Consequently, even if antibody titers significantly declined, a corresponding statistically significant drop in seropositivity might not always be detected. In contrast, Figure 2 used continuous antibody titer data, allowing for a more sensitive analysis that detected significant decreases at all time points. Figure 2 and Figure 3 were created from the same dataset. Figure 3 represented the stratified analysis of Figure 2 based on HLA matching. We have suggested to the editorial team to use a clearer image for better visualization. We sincerely hope this explanation is helpful.

Reviewer’s Comment: For Table 3, please provide the statistical analysis to determine if the differences were or were not significant.

Response: We sincerely appreciate the reviewer’s valuable suggestions. Table 3 presents the GMFRs between each time point. Specifically, M1-3/D0 represents the fold increase in antibody titers from D0 to M1-3 for each patient, with the geometric mean of these values reported as GMFR. For example, the GMFR of 74.91 for Diphtheria at M1-3/D0 indicates that antibody titers at M1-3 increased, on average, 74.91-fold compared to D0. Since each GMFR represents a within-patient fold change between two specific time points, direct comparisons between different GMFRs (e.g., M1-3/D0 vs. Y1/M1-3) may not provide meaningful insights. While this study has a small sample size and normality was not formally demonstrated, the GMFR (95% CIs) may provide valuable statistical insights. In general, when the 95% CI does not span 1, a statistically significant change can be inferred. However, this is an indirect method of assessing statistical significance rather than a formal hypothesis test. In this study, all results showed 95% CIs that did not span 1, suggesting statistical significance. That said, as the assumption of normality cannot be guaranteed, making definitive conclusions solely based on this method may not be entirely appropriate. To enhance clarity, we have added a brief explanation in the Results section (Line 256 in the revised manuscript).

Reviewer’s Comment: Why is the graphic of figure 2 different from figure 3?

Response: The primary difference between the two figures is whether the data was stratified by HLA-matched and mismatched cases.

Figure 2 presents antibody titers at three time points—pre-vaccination, 1–3 months after completing vaccination, and 1 year after vaccination—for all patients. Each graph represents a specific antigen, and statistical significance for changes between time points is indicated.

Figure 3, in contrast, displays the same data as Figure 2, but stratified by HLA-matched and mismatched cases. It also presents a statistical comparison assessing whether there were significant differences in antibody titers between the HLA-matched and mismatched groups at each time point.

We sincerely appreciate the reviewer’s insightful comment, which has helped us clarify this aspect of the manuscript.

Reviewer’s Comment: Why, if in figure 3 they employed a graphic that assumes a parametric distribution, did the authors perform a non-parametric test?

Response: We chose this graphical representation as we believed it would provide a clearer visualization of the data. The values are presented as GMTs, which is a widely accepted method in vaccinology and does not inherently assume a parametric distribution. Although the graphical format follows this standard approach, the actual statistical analysis was performed using a non-parametric test due to the small sample size and potential deviations from normality. This methodology aligns with previous studies that have used similar approaches for analyzing immunogenicity data. We appreciate your consideration of this approach.

Reviewer’s Comment: Also, data from time point 0 is related to point 1 and point 2, but the authors employed a statistical test to un-relate data (independent observation). This is not correct; they should employ a statistical test for dependent or related observations; they are the same patients at a different point in time. This applies for all the analysis.

Response: We sincerely apologize for any lack of clarity in our explanation. Figure 3 presents antibody titers at three time points—pre-vaccination, 1–3 months post-vaccination, and 1-year post-vaccination—stratified by HLA-matched and HLA-mismatched cases. To examine differences in antibody titers between the HLA-matched and mismatched groups at each individual time point, we employed the Mann-Whitney U test, as the two groups were considered independent at each time point. If the objective were to assess longitudinal changes in antibody titers within the same group over time, a dependent or paired statistical test, such as the Wilcoxon matched-pairs signed rank test, would indeed be more appropriate. However, since our primary analysis focuses on between-group comparisons at each time point rather than within-group changes over time, we believe that our current statistical approach is valid. That said, if we have misunderstood any aspect of the reviewer's concern, we would greatly appreciate further clarification. We have also improved the description in the Methods section (Line 155 in the revised manuscript).

Conclusion

We appreciate the reviewer’s detailed feedback, which has greatly improved our manuscript. We have carefully addressed all comments and made the necessary revisions accordingly. We hope that the revised manuscript meets your expectations and look forward to your further evaluation. We would greatly appreciate any further guidance on areas that may require additional refinement.

Sincerely,

Taiichiro Kobayashi, M.D., Ph.D. (taiichirou_kobayashi@tmhp.jp)

Department of Infectious Diseases

Tokyo Metropolitan Cancer and Infectious Diseases Center Komagome Hospital

3-18-22, Honkomagome, Bunkyo-ku, Tokyo 113-8677, Japan

Round 2

Reviewer 1 Report

Comments and Suggestions for Authors

Dear Authors,

The manuscript has been revised, and earlier issues have been corrected.
It may be published.

Author Response

Dear Reviewer,

Thank you very much for your kind message and for your valuable feedback during the review process. We sincerely appreciate your time and effort in evaluating our manuscript.

We are grateful for your understanding and support in the revisions, and we truly appreciate your guidance in improving the quality of our work.

Best regards,
Taiichiro Kobayashi

Reviewer 2 Report

Comments and Suggestions for Authors

1) About the response that they did not calculate the sample size, this is not acceptable. "We did not perform a sample size calculation as we did not have the financial resources to enroll a larger number of patients.". What if the sample size was small? Or if the sample size is bigger, that is the power of the sample and the significance of the results?

2) Please calculate the sample power.

3) This point is critical due to the p values in table 2. 

4) Also the authors did not performe several of the statistical request of the reviewer. 
